# DIFFUSION MODELS AS INTRINSIC DISTRIBUTION ESTIMATORS FOR SELF-VERIFYING INFERENCE-TIME SCALING

## ABSTRACT

To enhance sample quality beyond their standard outputs, diffusion models typically rely on inference-time scaling, a process that necessitates external verifiers. We challenge this dependency by proposing a framework that reframes the generative model itself as an intrinsic distribution estimator. Our framework provides the theoretical base and empirical evidence for this, showing that the distance between independent noise and diffusion model output serves as a proxy for a sample's distributional conformity. This insight enables our proposed method, Self-Verifying inference-time scaling method to directly assess at intermediate denoising step and to eliminate the need for external modules. Experiment results demonstrate that our scaling method achieves consistent improvements across diverse benchmarks in fidelity, preference, and compositionality. Our study establishes that the process of generating diffusion models is also an evaluative process, opening new avenues toward more resource-efficient and intrinsically aware generative models.

## 1 INTRODUCTION

Diffusion models have become a cornerstone of modern artificial intelligence, achieving state-of-the-art results across a multitude of domains. Their remarkable, modality agnostic expressive power has made them the mainstream choice for high-fidelity generation of images (Esser et al., 2024; Saharia et al., 2022; Betker et al., 2023), audio (Lee et al., 2025), and text (Li et al., 2022; Arriola et al., 2025). Beyond their generative prowess, the unique characteristics of diffusion models, such as their multi-step denoising process and their capacity for implicit likelihood estimation have enabled a range of intrinsic capabilities from zero-shot classification (Clark & Jaini, 2023; Li et al., 2023) and out-of-distribution (OOD) detection (Aithal et al., 2024; Zhang et al., 2025; Yao et al., 2024; Heng et al., 2024) to unsupervised image editing (Mokady et al., 2023; Huang et al., 2025).

A particularly promising application leveraging these properties is inference-time scaling, which aims to iteratively refine generation quality. However, prevailing methods typically rely on external pretrained reward model to evaluate a fully denoised sample Ma et al. (2025); Fernandes et al. (2025); Xie et al. (2025); Li et al. (2025). If the sample fails to meet certain criteria, the denoising process is reverted to an intermediate noisy state and corrected. This paradigm is inherently computationally expensive, as it requires at least one full denoising cycle for verification and introduces a dependency on auxiliary models.

This raises a fundamental question: can a diffusion model verify its own generation quality during the denoising process, without external supervision? In this work, we answer in the affirmative way by proposing a new perspective. We assume an objective as $\|a - M(f(a, b))\|^2$, where conditional generative model $M$ minimizes the objective, $a \sim \mathcal{N}, b \sim q_{data}$, and $f$ is linear combination function. We prove well-trained model $M^*$ can be viewed as an implicit distribution estimator and consequently distinguishing in-distribution (ID) and OOD samples, described in Eq. 5.

In addition, we demonstrate through reparameterization that this theoretical framework directly maps to the training objective of diffusion models. The model's noise prediction error at any given timestep serves as a proxy for the sample's conformity to the learned data distribution. We empirically validate this hypothesis using a pre-trained Stable Diffusion XL (SDXL) (Podell et al., 2023)

model, showing it can robustly differentiate ID ImageNet-1k (Deng et al., 2009) samples from OOD dataset like ImageNet-A (Hendrycks et al., 2021) and ImageNet-C (Hendrycks & Dietterich, 2019).

Building on this principle, we introduce a self-verifying inference-time scaling method that can estimate the reward at the middle of denoising steps with diffusion model itself without relying on the external verifier, by estimating which candidates has smaller distance to independent noise than the others. This approach leads to consistent performance improvements in unconditional generation on ImageNet and LSUN (Yu et al., 2016), human preference alignment in text-to-image synthesis, and compositional generation capabilities.

The main contributions of our work are as follows:

- We introduce a new framework that formalizes conditional noise generative models as implicit **distribution estimator**, capable of distinguishing ID from OOD components, by comparing between independent noise and model prediction.

- We demonstrate that this framework can be applied directly to diffusion models **theoretically** by reparameterization and **empirically** proven in ImageNet-1k, A, and C experiments.

- We propose a new inference-time scaling method that is **self-verifiable** and efficient, directly utilizing intermediate denoising sample to enhance generation quality **without requiring external verifiers or full generation steps**.

## 2 RELATED WORK

Diffusion models (Sohl-Dickstein et al., 2015; Ho et al., 2020) are generative model that learn to create new data by reversing a gradual noising process. This is accomplished by learning the reverse of the predefined forward process, which is structured as a Markov chain. The forward process, $q$, progressively adds Gaussian noise to the original data $x_0 \sim q(x_0)$ over $T$ timesteps. The transition at each timestep $t$ is defined as: $q(x_t|x_{t-1}) := \mathcal{N}(x_t; \sqrt{1-\beta_t}x_{t-1}, \beta_t \boldsymbol{I})$ where $\beta_t \in (0,1)$ are small positive constants defined by a variance schedule. A key property of this process is that we can sample $x_t$ at an arbitrary timestep $t$ directly from $x_0$ in a closed form. Letting $\alpha_t := 1 - \beta_t$ and $\bar{\alpha}_t := \Pi_{i=1}^t \alpha_t$, expressed as:

$$q(x_t|x_0) = \mathcal{N}(x_t; \sqrt{\bar{a}_t}x_0, (1-\bar{a}_t)\boldsymbol{I}), \quad x_t = \sqrt{\bar{\alpha}_t}x_0 + \sqrt{1-\bar{\alpha}_t}\epsilon \tag{1}$$

where $\epsilon \sim \mathcal{N}(0, \boldsymbol{I})$. The core of generation lies in learning the reverse process, $p_\theta$, which denoises the data starting from pure noise $x_T \sim \mathcal{N}(0, \boldsymbol{I})$ by iteratively sampling $x_{t-1} \sim p_\theta(x_{t-1}|x_t)$ until $x_0$ is produced. This process is modeled as $p_\theta(x_{t-1}|x_t) := \mathcal{N}(x_{t-1}; \mu_\theta(x_t, t), \Sigma_\theta(x_t, t))$ in the DDPM (Denoising Diffusion Probabilistic Models) (Ho et al., 2020). Instead of optimizing the variational lower bound (VLB) on log-likelihood, DDPM proposed a simplified objective that is proportional to the VLB. The model, $\epsilon_\theta(x_t, t)$, is trained to predict the noise component $\epsilon$ that was added to the data at timestep $t$. The objective function is:

$$L_{\text{simple}}(\theta) := \mathbb{E}_{t,x_0,\epsilon}\left[||\epsilon - \epsilon_\theta(\sqrt{\bar{\alpha}_t}x_0 + \sqrt{1-\bar{\alpha}_t}\epsilon, t)||^2\right] \tag{2}$$

This noise prediction network $\epsilon_\theta$ has predominantly been implemented using a U-Net (Ronneberger et al., 2015; Ho et al., 2020; Dhariwal & Nichol, 2021; Rombach et al., 2022). More recently, architectures like the DiT (Diffusion transformer) (Peebles & Xie, 2023; Esser et al., 2024) have replaced the U-Net backbone with a Transformer, demonstrating scalability.

Following the initial success of DDPMs, subsequent work has focused on improving sample quality, computational efficiency, and sampling speed. Sample fidelity was enhanced though techniques like guidance (Dhariwal & Nichol, 2021; Ho & Salimans, 2022; Karras et al., 2024), while Latent Diffusion Models (LDMs) (Rombach et al., 2022) drastically reduced computational costs by operating in a compressed latent space. To address the slow sampling speed, Denoising Diffusion Implicit Models (DDIMs) (Song et al., 2020) introduced a deterministic sampling process, and reformulating diffusion as an SDE/ODE enabled the use of fast numerical solvers like DPM-Solver (Lu et al., 2022). More recently, Consistency Models (Song et al., 2023; Luo et al., 2023) and Rectified Flow (Liu et al., 2022; 2023) have achieved generation in a single or very few steps without significant degradation by learning a direct noise-to-data mapping.

The ability of diffusion models to precisely model complex data distributions has also made them a powerful tool for applications beyond generation, such as anomaly detection (AD) or Out-of-Distribution (OOD) prediction. Current diffusion-based AD methods typically fall into two categories from Liu et al. (2025b) of reconstruction, density-based, and the other is the the flow-based approach. Reconstruction-based methods (Bercea et al., 2023; Zhang et al., 2025; Yao et al., 2024) leverage the principle that a model trained on normal data will yield higher reconstruction errors for anomalous inputs. In contrast, density-based approaches (Livernoche et al., 2023; Luo, 2023) use the learned distribution directly, often employing the magnitude of the score function or an estimated diffusion time as an anomaly score. Flow-based research (Heng et al., 2024; Aithal et al., 2024) utilizes the denoising trajectories to distinguish In-Distribution (ID) samples and OOD samples.

Finally, significant research has focused on aligning model outputs with human preferences and further enhancing image quality. One prominent approach involves fine-tuning the model directly on preference data or utilizing a reward model, techniques successful in language models. For instance, DPO-diffusion (Wallace et al., 2024) adapts Direct Preference Optimization (DPO) (Rafailov et al., 2023) and Flow-GRPO (Liu et al., 2025a) leverages Group Relative Policy Optimization (GRPO) (Shao et al., 2024) with Rectified Flow model to few-step denoising in order to achieve online-RL with an external verifier. Complementary to fine-tuning, another line of work explores inference-time scaling techniques. These methods typically generate multiple candidates and use an external verifier to select the optimal output Xie et al. (2025); Fernandes et al. (2025). For example, research has explored techniques such as Best-of-N sampling, Zero-Order Search, and Search over Paths, which compare fully denoised samples to identify the one that best aligns with desired criteria or quality metrics (Ma et al., 2025). This strategy of leveraging an external verifier to guide generation at inference time has become a common approach for scaling the performance of diffusion models.

## 3 IDENTIFYING OUT-OF-DISTRIBUTION VIA DIFFUSION MODEL AS A DISTRIBUTION ESTIMATOR

### 3.1 CONDITIONAL GENERATIVE MODEL AS A DISTRIBUTION ESTIMATOR

In this subsection, we investigate the robustness of the distribution estimation model to out-of-distribution (OOD) perturbations in a controlled generative setting. Our goal is to formalize and analyze how model performance degrades when a component of the input data is drawn from outside its training distribution.

**Generative Process and Learning Objective** Let $a \in \mathbb{R}^d$ be a latent variable representing a signal, drawn from a standard multivariate Gaussian distribution $a \sim \mathcal{A} = \mathcal{N}(0, \boldsymbol{I}_d)$. Let $b \in \mathbb{R}^d$ be a structured data component, drawn from an arbitrary high-dimensional data distribution $B$ with mean $\mu_b$ and covariance $\Sigma_b$. We assume a and b are statistically independent.

The model input $x \in \mathbb{R}^d$ is generated by a linear combination of these components, $x = f(a, b) = c_1 a + c_2 b$, where $0 \leq c_1, c_2 \leq 1$ are scalar coefficients controlling the relative influence of $a$ and $b$. We consider a model $M : \mathbb{R}^d \to \mathbb{R}^d$ that learns to produce samples following the conditional distribution of $a$ given $x$.

Instead of comparing the model output with the same ground-truth sample $a$, we introduce an independent draw $a' \sim \mathcal{A}$ to treat $M$ as a distributional estimator of $a$: a well-trained model should produce outputs whose distribution matches that of $a$. The optimal model $M^*$ minimizes the Mean Squared Error (MSE), which is equivalent to the statistical risk, with respect to this independent reference $a'$.

$$M^* = \underset{M}{argmin}\mathcal{L}(M), \quad \mathcal{L}(M) = \mathbb{E}_{a,b,a'}[\|a' - M(f(a,b))\|^2] \tag{3}$$

This risk decomposes as,

$$\mathbb{E}_{a,b,a'}[\|a' - M(f(a,b))\|^2] = \underbrace{\mathbb{E}\|a'\|^2}_{prior\ variance\ (constant)} + \mathbb{E}_{a,b}\|M(f(a,b))\|^2 \tag{4}$$

Since the first term is constant, minimizing $\mathcal{L}(M)$ is equivalent to matching the second moment of $M(f(a,b))$ to that of the true prior $\mathcal{A}$.

**In-Distribution (ID) vs Out-Of-Distribution (OOD) Risk** We now study model performance when faced with an outlier sample $b^*$ not representative of $\mathcal{B}$.

Such a $b^*$ may lie in a low probability region under $\mathcal{B}$ or originate from a different distribution. We define two key quantities. First, ID Risk, the expected loss achieved by the optimal model $M^*$ on data sampled from the training distribution: $R_{ID} = \mathbb{E}_{a,b,a'}[\|a' - M^*(c_1 a + c_2 b)\|^2]$ and second OOD Risk, the expected loss when $b$ is replaced by fixed outlier $b^*$: $R_{OOD}(b^*) = \mathbb{E}_{a,a'}[\|a' - M^*(c_1 a + c_2 b^*)\|^2]$.

Because the constant term $\mathbb{E}\|a'\|^2$ cancels in the comparison, the difference between $R_{ID}$ and $R_{OOD}(b^*)$ is fully determined by the change in the second moment of the model outputs.

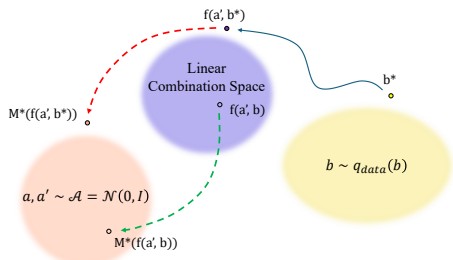

Figure 1: Visualization of the model's distribution estimation for ID and OOD data. The green dashed arrow shows that the optimal model $M^*$ successfully estimates the true distribution when conditioned on an ID sample $f(a', b)$. Conversely, the red dashed arrow illustrates the model's failure to estimate the distribution when conditioned on an OOD sample $f(a', b^*)$.

We hypothesize that a model optimized for the statistics of $\mathcal{B}$ will suffer a degradation in performance when evaluated on $b^*$, leading to:

$$R_{ID} \leq R_{OOD}(b^*)$$
$$\Rightarrow \mathbb{E}_{a,b,a'}[\|a' - M^*(c_1 a + c_2 b)\|^2] \leq \mathbb{E}_{a,a'}[\|a' - M^*(c_1 a + c_2 b^*)\|^2] \tag{5}$$

**Conditions for the Inequality** The inequality $R_{ID} \leq R_{OOD}(b^*)$ is guaranteed under the following sufficient conditions:

- Bayes-Optimal Model: $M^*$ exactly reproduces the true conditional distribution $M^*(x) \stackrel{d}{=} a|x$. In this case, the expected loss reduces to the sum of the prior variance and the conditional second moment of a. Replacing $b$ with $b^*$ induces a conditional distribution different from the training distribution, which can only increase (or leave unchanged) the expected second moment, yielding $R_{ID} \leq R_{OOD}(b^*)$.

- Well-Specified Generative Model: The assumed generative process $x = f(a, b)$ and prior $p(a) = \mathcal{N}(0, \boldsymbol{I}_d)$ match the true data-generating process. If the model is mis-specified or underfitted, there may exist $b^*$ such that $R_{OOD}(b^*) < R_{ID}$.

- Atypicality of $b^*$: The sample $b^*$ lies in a statistically atypical region of $B$, e.g., $p_B(b^*) \ll p_B(\mu_b)$. If $b^*$ is typical, then $R_{OOD}(b^*) \approx R_{ID}$.

Under these conditions, the OOD risk must be at least as large as the ID risk, with strict inequality whenever $p(a|x)$ under $b^*$ sufficiently deviates from the training-time conditional distribution.

## 3.2 DIFFUSION MODEL AS A DISTRIBUTION ESTIMATOR

Let the objective from our assumed model, $M$, be defined as:

$$L = \mathbb{E}_{t \sim p(t)}[\mathbb{E}_{a \sim \mathcal{N}(0,\boldsymbol{I}),b \sim q}[\|a - M(c_{1,t}a + c_{2,t}b, c_{1,t}, c_{2,t})\|^2]] \tag{6}$$

where $(c_{1,t}, c_{2,t})$ are coefficients indexed by a timestep t drawn from a distribution $p(t)$.

We begin by establishing a set of one-to-one correspondences to align our proposed circumstances with the DDPM framework. First, we map the variable $a \leftrightarrow \epsilon$ and $b \leftrightarrow x_0$, where both $a$ and $\epsilon$ are sampled from $\mathcal{N}(0, \boldsymbol{I})$, and both $b$ and $x_0$ are from the data distribution $q$. Second, the coefficients $(c_{1,t}, c_{2,t})$ are defined to match the DDPM schedule: $c_{1,t} := \sqrt{1 - \bar{\alpha}_t}$ and $c_{2,t} := \sqrt{\bar{\alpha}_t}$. Given that the timestep $t$ uniquely determines these coefficients, our model $M$ (conditioned on coefficients) becomes functionally identical to the DDPM model $\epsilon_\theta$ (conditioned on $t$). This equivalence is formally expressed as $M(\cdot, c_{1,t}, c_{2,t}) \equiv \epsilon_\theta(\cdot, t)$.

By substituting these mappings directly into our redefined objective (Eq. 6), we obtain:

$$L = \mathbb{E}_{t \sim p(t)}[\mathbb{E}_{\epsilon \sim \mathcal{N}(0,\boldsymbol{I}),x_0 \sim q}[\|\epsilon - \epsilon_\theta(\sqrt{\bar{\alpha}_t}x_0 + \sqrt{1 - \bar{\alpha}_t}\epsilon, t)\|^2]] \tag{7}$$

This resulting expression is identical in form and substance to the DDPM objective (Eq. 2). Therefore, we have formally shown that $L = L_{DDPM}$. The proposed framework is an exact reparameterization of the DDPM noise-prediction training paradigm.

### 3.3 IS YOUR DIFFUSION MODEL REALLY DISTRIBUTION ESTIMATOR?

To empirically validate our theoretical framework (Eq. 5), we conduct experiments to determine if a pre-trained diffusion model can effectively distinguish between in-distribution (ID) and out-of-distribution (OOD) data by measuring noise prediction discrepancies.

**Distribution Estimation Setup** We employ Stable Diffusion XL (SDXL) (Podell et al., 2023), a publicly available latent diffusion model, as our off-the-shelf distribution oracle. For our in-distribution data, we use the validation set of ImageNet-1k (Deng et al., 2009), which represents the model's learned data manifold. To challenge the model with out-of-distribution samples, we leverage two datasets: ImageNet-A (Hendrycks et al., 2021), a curated dataset of ImageNet-1k that are naturally adversarial to classifiers, and ImageNet-C (Hendrycks & Dietterich, 2019), a dataset for evaluating robustness against common visual corruptions.

Based on our assumption, the diffusion model will exhibit a higher noise prediction error for OOD samples than for ID samples. Given a pair of images, we first provide the model with a prompt `"a photo of {class_name}"`, corresponding to the image's ground-truth label. Next, we compute the L2 distance between the independent noise $\epsilon'$ and the model's predicted noise $\epsilon_\theta(x_t(x, t, \epsilon))$ at several distinct timesteps, where $\epsilon, \epsilon'$ are independently sampled from the Gaussian distribution. This "noise distance" serves as a proxy for how well the image conforms to the model's learned distribution. Finally, we classify the image with the lower average noise distance across these timesteps as the in-distribution sample. A prediction is deemed correct if the model successfully identifies the ImageNet-1k sample(ID).

For the ImageNet-A comparison, we randomly sample 10 pairs of images for each of ImageNet-A's 200 classes, comparing each ImageNet-A sample against a randomly selected ImageNet-1k sample from the same class. For the ImageNet-C evaluation, we compare each corrupted image against its original, clean counterpart from ImageNet-1k, using 5 such pairs for every class. Due to computational constraints, we limit out ImageNet-C analysis to four corruption types (defocus blur, contrast, elastic transform, saturate) at severity levels 1 and 3.

**Distribution Estimation Results** Our experiments confirm that the noise prediction error of SDXL is a reliable indicator for distinguishing ID from OOD data.

| ImageNet-A | ImageNet-C | | | | | | | |
| --- | --- | --- | --- | --- | --- | --- | --- | --- |
| | defocus | | contrast | | elastic transform | | saturate | |
| | 1 | 3 | 1 | 3 | 1 | 3 | 1 | 3 |
| 56.5% | 98.6% | 99.3% | 98.4% | 99.6% | 92.8% | 89.7% | 89.3% | 80.7% |

Table 1: Accuracy of our OOD detection method on subsets of the ImageNet-A and ImageNet-C dataset. The model classifies which image in a pair is ID based on noise prediction error.

When tasked with discriminating between ImageNet-1k and the challenging, naturally adversarial samples of ImageNet-A, our method achieved an accuracy of **56.5%**. While modest, the result is significantly above 50% of random choice, demonstrating that the model can perceive near-OOD images, even though these OOD images came from the ID set.

The model's discriminative power was substantially more pronounced on the ImageNet-C dataset, as detailed in Table 1. As the results show, the model achieves near-perfect accuracy on simpler corruptions like defocus and contrast. Interestingly, for structurally complex corruptions such as elastic transform and saturate, the accuracy remains high but shows a slight decrease as severity intensifies. A plausible explanation is that the severe artifacts introduce high-frequency distortions that statistically mimic the Gaussian noise the model is trained to predict, making the OOD samples harder to distinguish from their ID counterparts.

---

**Algorithm 1** Self-Verifying Inference-Time Scaling

---

**Require:** Diffusion model $\epsilon_\theta$, denoising timestep set $t = \{T..1\}$, normal dist. $\mathcal{N}$, non-deterministic scheduler $S$, inference-time scaling factor at its corresponding timestep $n_t$.

1: Sample $\epsilon = \{\epsilon_1, \ldots, \epsilon_{n_T}\}$ from $\mathcal{N}$           ▷ Sample $n$ noise vectors
2: **for** $i = 1 \rightarrow n_T$ **do**
3:      $x_i \leftarrow \epsilon_i$           ▷ Initialization
4: **end for**
5: **for** $t$ in $t$ **do**
6:      **for** $i = 1 \rightarrow n_t$ **do**
7:          noise_dist$_i \leftarrow \|\epsilon_\theta(x_{i,t}(\epsilon, t), t) - \epsilon'\|^2$           ▷ where $\epsilon, \epsilon' \sim \mathcal{N}$
8:      **end for**
9:      $i_{\text{best}} \leftarrow argmin_i(\text{noise\_dist}_i)$           ▷ Choose best sample
10:      $x_t \leftarrow x_{i_{\text{best}},t}$
11:      **for** $i = 1 \rightarrow n_t$ **do**
12:          $x_{i,t-1} \leftarrow S.\text{step}(\epsilon_\theta(x_t, t), x_t, t)$           ▷ Denoise $n$ times in parallel
13:      **end for**
14: **end for**
15: **return** $x_0$

---

# 4 INFERENCE TIME SCALING THROUGH DISTRIBUTION ESTIMATION OF THE DIFFUSION MODEL

The remarkable performance of large language models (LLMs) has been significantly enhanced by inference-time strategies (Muennighoff et al., 2025) that expand the search space for solutions, such as chain-of-thought prompting (Kojima et al., 2022) and self-consistency (Bartsch et al., 2023), followed by a verification or selection step (Yao et al., 2023; Liu et al., 2025c). These methods leverage the model's extensive knowledge to generate and evaluate multiple candidate outputs, ultimately improving final performance without costly retaining (Snell et al., 2024).

Current inference-time scaling approaches for diffusion models often employ an external verifier to score and select the best candidates. This approach typically requires intermediate samples to be fully denoised into a clean image at each evaluation step, consequently costs intensive.

We propose a reasonable approach that circumvents these complexities. We have shown that the diffusion model can be a distribution estimator (Eq. 5), hypothetically and empirically. Building on this following, we further posit that this inherent capability can be repurposed as an internal quality verifier, allowing the model to assess the plausibility at its intermediate generative trajectories, by comparing the noise distance of the denoising candidates.

**Self-Verifying Inference-Time Scaling for Diffusion Model** Our algorithm, Self-Verifying Inference-Time Scaling, operationalizes this principle through a step-wise choice and branching procedure. The process begins with setting a number of candidates at timestep $n_t$ and samples $n_T$ noises from a Gaussian distribution. At each timestep $t$ from $T$ down to 1, we evaluate all current candidates using verification score, compute independent noise distance with model predict noise (line 7 from Algorithm.1). The single candidate with the lowest noise distance is selected, and all other are pruned. A stochastic denoising scheduler is then used to branch from this single best candidate, generating a diverse set of $n_t$ candidates for the subsequent step. This iterative cycle of verifying, selecting, and branching efficiently guides the generation process along the most self-preferred trajectory to produce the final image.

# 5 EXPERIMENTS

We conduct a comprehensive set of experiments to validate the effectiveness of our proposed method, which is called Self-Verify. We evaluate its performance against a vanilla baseline on both unconditional and text-to-image generation tasks across a variety of benchmarks. Furthermore, we perform a detailed ablation study to analyze the impact of different hyperparameter choices and to identify an efficient configuration for our method.

## 5.1 EXPERIMENTAL SETUP

Our method, Self-Verify, is compared against a standard vanilla sampling baseline. Based on our ablation studies (detailed in Section 5.2), we adopt an efficient default configuration for Self-Verify: we maintain 4 candidates for the first four timesteps and then prune to a single trajectory for the remaining steps. This results in a modest increase in the number of function evaluations (NFE) from 50 for the vanilla sampler to 62 for Self-Verify.

**FID evaluation** For unconditional image generation, we evaluate Fréchet Inception Distance (FID) (Heusel et al., 2017) and Inception score (IS) (Salimans et al., 2016). We use the pre-trained ADM models (Dhariwal & Nichol, 2021) and their official checkpoints on the ImageNet 256x256 (Deng et al., 2009), LSUN-Cat 256x256, and LSUN-Bedroom 256x256 (Yu et al., 2016) datasets. Following standard evaluation protocol, we use the DDIM (Song et al., 2020) sampler with 50 steps and $\eta = 1$. We generate 10,000 samples for each experiment and compute FID and IS scores against the pre-computed reference statistics.

**Text-to-Image Human Preference Evaluation** To assess performance on text-to-image generation, we use the Stable Diffusion XL (Podell et al., 2023) model with the DPMSolverSDE (Lu et al., 2022) sampler. We use a set of 500 prompts from the test split of the Pick-a-Pic dataset (Kirstain et al., 2023), which is designed for modeling human preferences. We evaluate the generated images using a suite of automated scoring models: CLIPscore (Hessel et al., 2022) for image-text alignment, and PickScore (Kirstain et al., 2023), ImageReward (Xu et al., 2023), HPSv2 (Wu et al., 2023), and Aesthetic score, which serve as strong proxies for human aesthetic and compositional preferences.

**Object-Oriented Evaluation** To measure the model's ability to follow compositional instructions, we utilize the GenEval (Ghosh et al., 2023). This benchmark evaluates performance across several fine-grained categories, including the rendering of colors, object count, spatial position, and color attribute. Experiments are conducted with the SDXL model and DPMSolverSDE sampler.

**Ablation** We conduct an ablation study to determine the most effective and efficient candidate scheduling function, $n_t$. This study uses the same environment as the text-to-image human preference evaluation. We investigate several strategies: (1) Constant Candidates: maintaining a constant number of candidates throughout, in this experiment 4, (2) maintaining $n$ candidates for a fixed number of initial timestep $t$ before reverting to vanilla sampling, and (3) dynamically decreasing the number of candidates from 16 to 1 using linear and exponential pruning schedules. We report the HPS score as the primary metric, with PickScore and ImageReward available in the Appendix A.2

## 5.2 EXPERIMENTAL RESULTS

**Main Results** Our experiments imply that Self-Verify shows consistent improvement in the quality of generated samples across a range of benchmarks and tasks. While the improvements sometimes modest, they demonstrate a systematic positive effect, validating the efficacy of our approach.

| Method | ImageNet | | LSUN-Cat | | LSUN-Bedroom | |
|---|---|---|---|---|---|---|
| | **FID** $\downarrow$ | **IS** $\uparrow$ | **FID** $\downarrow$ | **IS** $\uparrow$ | **FID** $\downarrow$ | **IS** $\uparrow$ |
| Vanilla | 20.73 | 86.346 | 19.89 | **4.974** | 8.54 | 2.368 |
| Self-Verify | **20.59** | **87.510** | **19.66** | 4.957 | **8.45** | **2.373** |

Table 2: Unconditional generation results. Self-Verify improves FID scores across all datasets with marginal compute overhead.

For **unconditional generation**, we observe a consistent trend of improved FID scores across the ImageNet, LSUN-Cat, and LSUN-Bedroom datasets (Table 2). This suggests that by pruning less plausible paths, our method guides the sampling process along a trajectory that better aligns with the learned data distribution, leading to enhanced overall sample fidelity.

In the **text-to-image domain**, Self-Verify systemically leads to systemically higher scores on metics that serve as strong proxies for human preference, including PickScore, ImageReward, and HPS (Table 3). It is particularly compelling that this enhancement is achieved without guidance from any external, human-aligned reward models during the inference process. We hypothesize this stems from the model's learned distribution implicitly encoding human aesthetic biases, akin to a mere exposure effect (Zajonc, 2001). High-quality and aesthetically pleasing im-

| Method | CLIP | PickScore | ImageReward | HPS | Aesthetic |
|---|---|---|---|---|---|
| Vanilla | 8.71 | 22.02 | 0.8361 | 0.2906 | 5.8928 |
| Self-Verify (4/4) | **28.78** | **22.12** | **0.8972** | 0.2935 | 5.9329 |
| Self-Verify (8/1) | 28.70 | 22.10 | 0.8929 | **0.2936** | **5.9732** |

Table 3: Text-to-image human preference evaluation on prompts from the test split of the Pick-a-pic dataset. Self-verify (n/t) implies maintaining $n$ candidates for a fixed number of initial timesteps $t$. Our method achieves the best performance across human preference metrics.

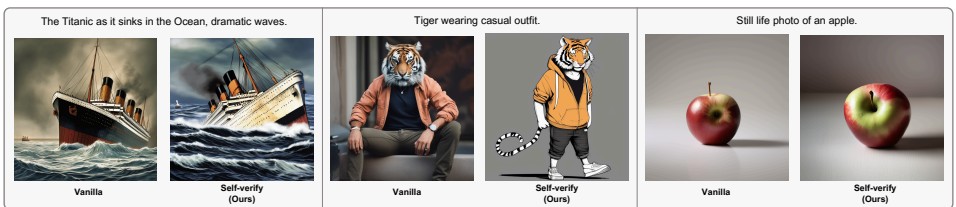

Figure 2: **Qualitative Comparison on Text-to-image domain.** Our approach yields outputs with stronger alignment to the textual prompt and higher aesthetic quality, whereas vanilla generations miss key prompt details or appear less coherent.

ages are more prevalent in the training data than flawed ones. Consequently, the model naturally learns a probabilistic distribution where desired samples are assigned higher density than undesired ones ($p(x_{undesired}) < p(x_{desired})$). By seeking the most self-consistent generative trajectory, our method is inherently guided toward these high-density, high-preference regions. The qualitative comparison in Figure 2 highlights the effectiveness of our method across three representative prompts. These examples show that Self-Verify yields more visually consistent generations.

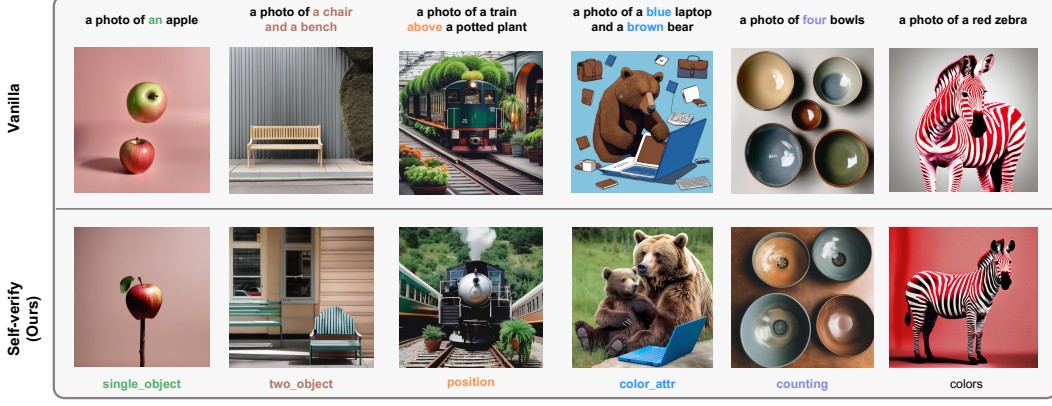

Figure 3: **Qualitative comparisons on the GenEval benchmark.** Compared to vanilla generations, Self-Verify (Ours) more faithfully follows prompt specifications across categories, leading to semantically accurate and coherent outputs.

| Category | Vanilla | Self-Verify |
|---|---|---|
| single_object | 97.81% | **98.44%** |
| two_object | 75.76% | **78.28%** |
| position | 11.00% | **11.25%** |
| color_attr | 18.25% | **20.50%** |
| counting | 39.38% | **41.88%** |
| colors | 84.31% | 84.31% |
| Overall score | 0.5442 | **0.5578** |

Table 4: Object-oriented evaluation on GenEval benchmark. Self-Verify consistently improves compositional correctness.

Furthermore, when assessing compositional understanding with the **GenEval benchmark**, our method achieves a higher overall score (Table 4). The gains are consistent for all categories. This indicates that the internal verification process helps the model better adhere to intricate prompt details, resulting in more semantically accurate and coherent images. Self-Verify consistently improves compositional understanding by adhering to object count, position, color attributes, and counting, while vanilla generations fail on these fine-grained details (Figure 3).

| HPS | | Candidates | | | | |
|---|---|---|---|---|---|---|
| | | 1 | 2 | 4 | 8 | 16 |
| Timesteps | 1 | 0.2906 (vanilla) | 0.2904 | 0.2915 | 0.2936 | 0.2929 |
| | 2 | - | 0.2915 | 0.2916 | 0.2918 | 0.2925 |
| | 4 | - | 0.2904 | 0.2935 | 0.2920 | 0.2917 |
| | 8 | - | 0.2893 | 0.2929 | 0.2930 | 0.2933 |
| | 16 | - | 0.2913 | 0.2911 | 0.2928 | 0.2940 |
| Constant Candidates | | 0.2921 | | | | |
| Linear Pruning | | 0.2923 | | | | |
| Exponential Pruning | | 0.2927 | | | | |

Table 5: Ablation study on candidate scheduling, reporting HPS score. The setting with 4 candidates for the first 4 steps offers a strong performance-cost trade-off. Constant candidates, Linear Pruning, and Exponential Pruning are described at Sec. 5.1

**Ablation Study**   Our ablation study on the candidate scheduling function provides insight into the method's cost-performance trade-off. The results confirm that applying verification for even a few initial steps yields sufficient performance gain over the vanilla baseline (Table 5). We found that the relationship between quality and computational cost is not linear; simply increasing the number of candidates or verification steps does not guarantee proportionally better results. The configurations chosen for our main experiments, which were 4 candidates for the first 4 steps and 8 candidates for the first step, were identified as an effective sweet spot. Additionally, the adaptive pruning results (below 3 rows) tell us these strategies cannot be strong substitutes.

# 6 LIMITATIONS AND FUTURE WORK

We acknowledge two primary limitations. First, our empirical validations of the suggested assumption and proposed method are confined to the visual domain. Validating the broader applicability of our framework is thus a critical next step. Investigating whether distribution estimation and its subsequent inference-time scaling holds for diffusion models in other modalities, such as audio synthesis and language modeling, will be essential to ascertain the domain generalizability of our approach.

Furthermore, the performance gains afforded by our method, while consistent, are incremental rather than transformative compare to approaches using an external verifier. A promising direction is therefore to bridge this performance gap, potentially by developing more nuanced aggregation methods for the intrinsic noise signal or by utilizing external verifiers to combine the schemes to amplify the model's capacity.

# 7 CONCLUSION

In this work, we present a new framework for interpreting conditional generative models as implicit distribution estimators (Sec. 3.1). Through the reparameterization of conditional generative model, we established that the diffusion model can serve this role, using their noise prediction to estimate the likelihood that a sample belongs to the learned data distribution (Sec. 3.2). Our initial experiments empirically confirmed this hypothesis, showing that a standard pre-trained diffusion model can effectively distinguish between in-distribution and out-of-distribution samples (Sec. 3.3).

Building on this principle, we proposed an inference-time scaling methodology that leverages the diffusion model itself as an intrinsic quality verifier (Sec. 4). This self-verifying approach adjusts the generation process without relying on any external reward models. Our comprehensive experiments demonstrated that this method yields consistent improvements across a range of tasks, enhancing unconditional generation fidelity on ImageNet and LSUN, improving human preference scores in text-to-image synthesis, and boosting compositional capabilities (Sec. 5.2).

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

# A APPENDIX

## A.1 LLM USAGE

We utilize LLMs, Gemini-2.5-pro and ChatGPT, to polish our writing and to get help experimental code implementation.

## A.2 MORE RESULTS FROM ABLATION STUDY

The results from the ablation study in Sec.5.2 on PickScore (Kirstain et al., 2023) from table 6 and ImageReward (Xu et al., 2023) from table. 7. The findings show a similar trend to the results from HPS (Table.5). Consequently, we selected the *"4 candidates from first 4 timesteps"* approach. For the text-to-image human preference alignment experiments, we also included the results for *"8 candidates for only the first timestep"*.

| PickScore | | Candidates | | | | |
|---|---|---|---|---|---|---|
| | | 1 | 2 | 4 | 8 | 16 |
| Timesteps | 1 | 22.02 (vanilla) | 22.04 | 22.07 | 22.08 | 22.03 |
| | 2 | - | 22.04 | 22.06 | 22.08 | 22.07 |
| | 4 | - | 22.02 | 22.12 | 22.06 | 22.06 |
| | 8 | - | 22.04 | 22.04 | 22.10 | 22.09 |
| | 16 | - | 22.04 | 22.07 | 22.06 | 22.13 |
| Constant Candidates | | 22.05 | | | | |
| Linear Pruning | | 22.05 | | | | |
| Exponential Pruning | | 22.07 | | | | |

Table 6: Ablation study on candidate scheduling, report PickScore score. Constant candidates, Linear Pruning, and Exponential Pruning are described at Sec. 5.1

| ImageReward | | Candidates | | | | |
|---|---|---|---|---|---|---|
| | | 1 | 2 | 4 | 8 | 16 |
| Timesteps | 1 | 0.8361 (vanilla) | 0.8543 | 0.8707 | 0.8929 | 0.8985 |
| | 2 | - | 0.8490 | 0.8389 | 0.8706 | 0.8600 |
| | 4 | - | 0.8356 | 0.8972 | 0.8681 | 0.8541 |
| | 8 | - | 0.8253 | 0.9201 | 0.8660 | 0.9024 |
| | 16 | - | 0.8619 | 0.8619 | 0.8516 | 0.8795 |
| Constant Candidates | | 0.8724 | | | | |
| Linear Pruning | | 0.8746 | | | | |
| Exponential Pruning | | 0.8908 | | | | |

Table 7: Ablation study on candidate scheduling, report ImageReward score. Constant candidates, Linear Pruning, and Exponential Pruning are described at Sec. 5.1

A.3   SAMPLES FROM UNCONDITIONAL GENERATION

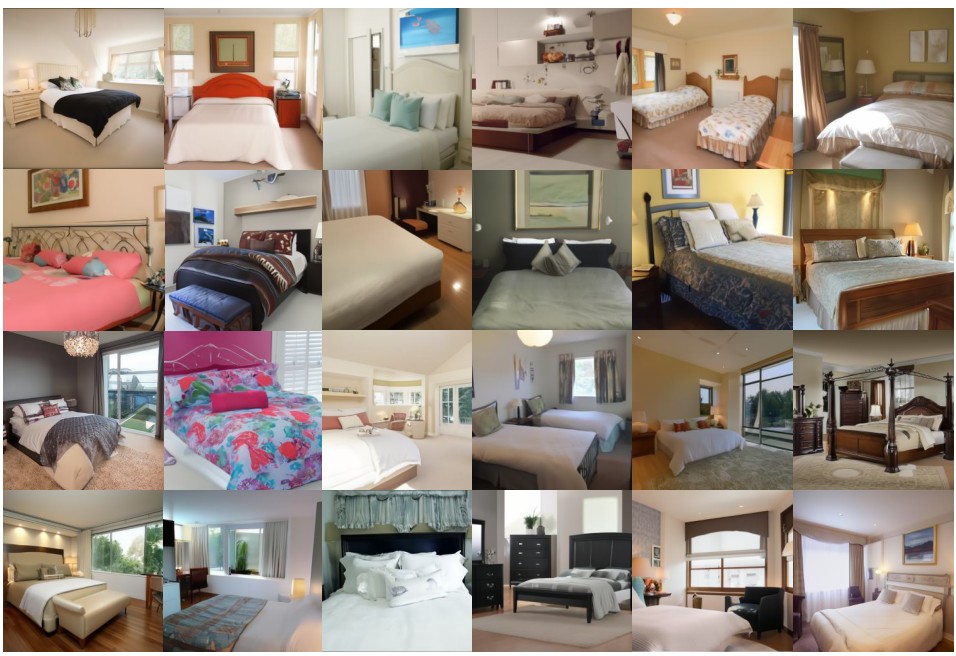

Figure 4: Unconditionally generated samples from model trained on LSUN-Bedroom 256x256 with our sampling methods. FID = 8.45

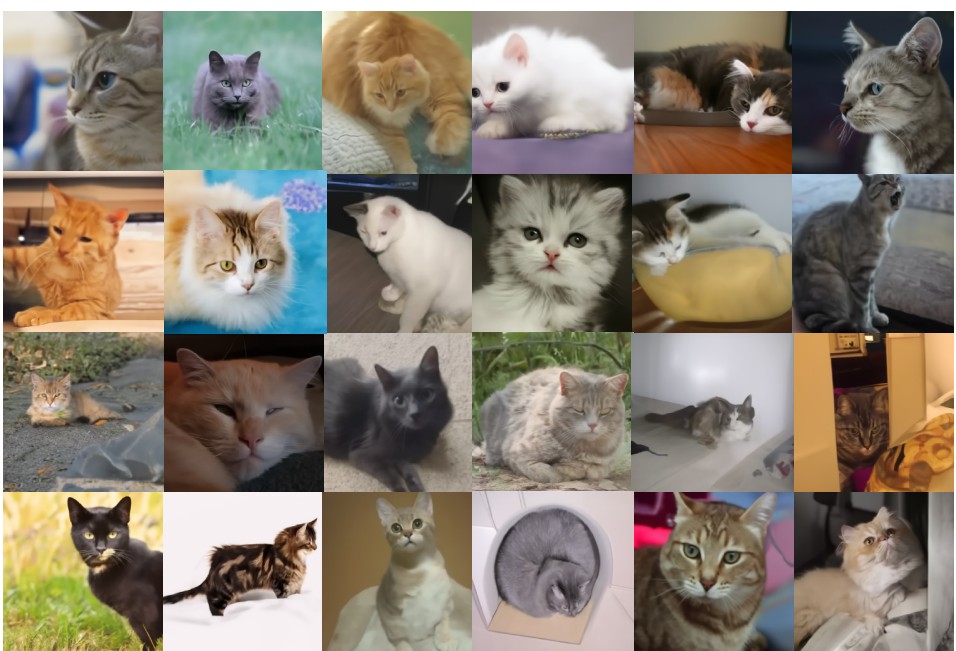

Figure 5: Unconditionally generated samples from model trained on LSUN-Cat 256x256 with our sampling methods. FID = 19.66

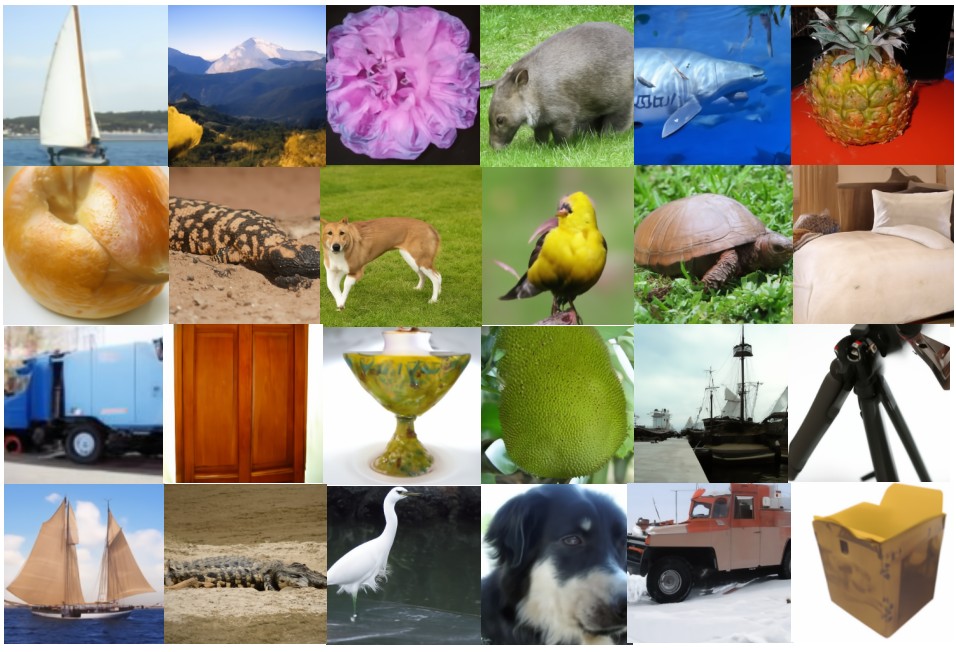

Figure 6: Unconditional generated samples from model trained on ImageNet 256x256 with our sampling methods. FID = 20.59

