# OpenReview forum: "Diffusion Models as Intrinsic Distribution Estimators for Self-Verifying Inference-Time Scaling"
_ICLR.cc/2026/Conference — ICLR 2026 Conference Withdrawn Submission_

### Official Review · Reviewer_D3kj · 2025-10-30

**Soundness:** 2
**Presentation:** 2
**Contribution:** 2
**Rating:** 4
**Confidence:** 3

**Summary:**

The paper proposes to reinterpret a diffusion model as an intrinsic distribution estimator by measuring the distance between the model’s predicted noise and an independently sampled Gaussian noise at intermediate timesteps. Under this view, the authors claim that a well-trained model should yield smaller “noise distance” for in-distribution (ID) inputs than out-of-distribution (OOD) ones, formalized via an inequality under strong sufficiency assumptions (Bayes-optimality, well-specified generative process, and atypicality of the OOD sample). They further show a one-to-one reparameterization that recovers the standard DDPM training objective and then leverage the same “noise distance” as a self-verifying inference-time scaling score that selects among multiple partial trajectories during sampling, without external reward models. Empirically, they (i) report limited OOD discrimination with SDXL (56.5% on ImageNet-A; near-perfect on several ImageNet-C corruptions), and (ii) demonstrate small but consistent gains in unconditional FID/IS on ImageNet/LSUN and in text-to-image preference proxies and GenEval compositionality, with modest extra compute (NFE 50→62). Overall, the method is simple, training-free, and appears to improve quality slightly across tasks, but theoretical novelty is limited (a reparameterization of DDPM loss), and experiments miss compute-matched and score-norm baselines.

**Strengths:**

- **Simplicity & zero-tuning**: The approach uses the model’s own intermediate signals; no auxiliary verifier or fine-tuning is required, which is attractive for deployment.
- **Coherent narrative**: The paper ties a (re)interpretation of the DDPM objective to a practical inference-time selection rule and provides an algorithmic realization (Algorithm 1).
- **Broad albeit modest empirical gains**: Consistent improvements in unconditional FID and preference proxies across datasets/prompts with modest added NFE (50→62). This suggests the signal is useful even in a greedy selection regime.
- **Ablations on branching schedules**: The study explores candidate counts/timesteps and identifies a reasonable quality-cost sweet spot (“4 candidates for first 4 steps”).

**Weaknesses:**

- **Theory adds limited novelty and relies on strong assumptions**: The ID≤OOD inequality is contingent on Bayes-optimality and well-specified modeling; otherwise it may not hold. The “distribution estimator” view largely restates the standard DDPM MSE in a different guise. More rigorous analysis (e.g., linking the proposed score to likelihood, score-matching identities, or calibrated risk bounds) is needed.
- **The proposed metric likely collapses to minimizing the predicted-noise norm**: Since ( \mathbb{E}\lVert \hat\epsilon - \epsilon'\rVert^2 = \mathbb{E}\lVert \hat\epsilon\rVert^2 + \mathbb{E}\lVert \epsilon'\rVert^2 ) with independence and zero mean, selecting the smallest “distance to independent noise” is nearly equivalent to selecting the smallest predicted-noise magnitude. The paper does not compare against this simpler proxy nor against known score-norm/energy baselines used for OOD/anomaly with diffusion models. (This is crucial to establish that the *independent-noise* variant offers unique value beyond known proxies.)
- **Missing compute-matched baselines**: Self-Verify uses 62 NFEs vs. 50 for vanilla. It is unclear whether equal-compute vanilla (e.g., 62-step sampler, more restarts, or Best-of-N at the end) would close the small FID/HPS gaps.
- **OOD evaluation is weak**: Only 56.5% on ImageNet-A (barely above chance), yet near-perfect on easy ImageNet-C corruptions, potentially measuring robustness to degradations rather than true semantic OOD. Protocol lacks details (how many timesteps scored, normalization across t, latent vs pixel spaces, classifier prompts), hindering reproducibility and interpretation.
- **Inconsistencies and missing statistical analysis**: CLIPScore jumps from 8.71 to ~28.7 suggests a scale mismatch or implementation bug; GenEval overall improvement is marginal (0.5442→0.5578) with some categories unchanged; no confidence intervals or significance tests are provided.
- **Comparison to prior inference-time scaling is incomplete**: The paper cites external-verifier methods but does not include direct comparisons under matched compute, leaving unclear how competitive the intrinsic verifier is.
- **Scope limited to vision**: Limitations admit domain restriction; given the strong framing, broader modality tests would strengthen the claim.

**Questions:**

1. **Compute-matching**: Please provide 62-NFE vanilla baselines (and 50 vs 62 step curves) and simple multi-seed Best-of-N at the *final* step to isolate the benefit of mid-trajectory verification.
2. **Noise-distance vs. norm/score baselines**: Compare your selection score to (a) (\lVert \hat\epsilon\rVert), (b) score-norm-based density proxies, and (c) estimated likelihood/consistency energy, to demonstrate that the *independent-noise* subtraction matters.
3. **CLIPScore scale & metrics**: Clarify CLIPScore computation pipeline (text/image preprocessing, normalization, range). Why is vanilla 8.71 while your variants ~28.7? Provide code or formal definition.
4. **OOD protocol details**: Which timesteps are used to average “noise distance”? How many samples per class/prompts? Are distances computed in latent or pixel space for SDXL? Please add sensitivity to the number of evaluated timesteps.
5. **Myopic selection**: Algorithm 1 greedily keeps a single best candidate each step. What happens with beam-k selection (retain top-k across steps) under equal compute? Any evidence of mode-dropping or loss of diversity?
6. **Ablation significance**: Add error bars/CI over multiple prompt shards for HPS/PickScore/ImageReward and GenEval; current differences are small and may fall within noise.
7. **Link to likelihood**: Can you derive a principled connection from your distance to (approximate) negative log-likelihood or to the score-matching objective, beyond reparameterizing the DDPM loss?

---

### Official Review · Reviewer_APdP · 2025-10-30

**Soundness:** 3
**Presentation:** 2
**Contribution:** 2
**Rating:** 4
**Confidence:** 4

**Summary:**

This paper introduces Self-Verify, a diffusion sampling framework that enhances output quality through inference-time scaling without external models. It first establishes trained conditional generative models as implicit distribution estimators.

Building on this, the method evaluates multiple candidates during intermediate denoising steps via a beam search-like iterative process, demonstrating strong performance on human preference metrics including HPS and PickScore.

**Strengths:**

1. The contribution of this work is an inference enhancement method that requires no external components, leveraging the conditional diffusion model's intrinsic out-of-distribution detection capability to optimize denoising trajectories, demonstrating notable effectiveness.

2. This paper demonstrates strong writing quality and a well-structured storyline. It first establishes diffusion models as intrinsic distribution estimators, then builds upon this foundation to develop the Self-Verify framework, supported by comprehensive charts and diagrams.

**Weaknesses:**

1. I think the authors should incorporate a more thorough discussion of related work. For instance, in the section claiming "Diffusion Models as Distribution Estimators (Sec 3.3)," this concept does not represent a novel discovery, as several prior works [1,2] have reached remarkably similar conclusions (despite the brief mention of Diffusion Classifier in the Introduction). It is recommended that the authors clarify the specific contributions of their theoretical derivation and explicitly distinguish their work from these existing studies.

2. Moreover, regarding the self-verify inference-scaling approach, numerous comparable studies [3,4] have explored enhancing sampling performance by searching for optimal noise/trajectories. Some of these methods employ external plugins (e.g., MLLMs), while others do not. However, this paper only compares against vanilla baselines in its experiments, which I consider to be an incomplete evaluation.

3. In the experimental section, relying solely on SDXL fails to demonstrate the method's generalizability. For instance, within the DiT architecture (e.g., Flux, SD3.5), it remains unclear whether Self-Verify would perform equally effectively.

[1] Li, Alexander C., et al. "Your diffusion model is secretly a zero-shot classifier." Proceedings of the IEEE/CVF International Conference on Computer Vision. 2023.

[2] Chen, Huanran, et al. "Your diffusion model is secretly a certifiably robust classifier." arXiv preprint arXiv:2402.02316 (2024).

[3] Oshima, Yuta, et al. "Inference-time text-to-video alignment with diffusion latent beam search." arXiv preprint arXiv:2501.19252 (2025).

[4] Ma, Nanye, et al. "Inference-time scaling for diffusion models beyond scaling denoising steps." arXiv preprint arXiv:2501.09732 (2025).

**Questions:**

1. How does this theoretical framework fundamentally differ from existing distribution estimation methods? Under what conditions is your noise distance metric superior to likelihood-based approaches?

2. To what extent does Self-Verify generalize to other diffusion architectures (e.g., DiT) and their conditional generation paradigms?

3. Does the paper overestimate the universality of using "**noise distance as a distribution proxy**"? How can the authors rule out the possibility that high noise distance merely reflects distribution shift rather than low quality?

---

### Official Review · Reviewer_SBaZ · 2025-10-30

**Soundness:** 2
**Presentation:** 3
**Contribution:** 3
**Rating:** 4
**Confidence:** 3

**Summary:**

This paper proposes that diffusion models can intrinsically estimate sample quality by using the distance between their noise predictions and an independent noise source as a measure of distributional conformity. Based on this principle, the authors introduce a self-verifying inference-time scaling method that evaluates and selects the most promising generation paths. Empirical results demonstrate that this approach consistently improves sample fidelity, human preference alignment, and compositional correctness in text-to-image and unconditional generation tasks.

**Strengths:**

- This paper presents a novel and insightful perspective that the model's internal noise prediction error during its own denoising process is a powerful, intrinsic metric for sample quality.
- The authors establish a ​theoretical foundation​ by reparameterizing the diffusion training objective to show its equivalence to a distribution estimator, providing a principled basis for their idea. This is then supported by ​empirical validation​ on established OOD detection tasks. They finally translate this insight into a ​practical algorithm​ (Self-Verify) and conduct comprehensive experiments across unconditional generation, human preference, and compositionality benchmarks.
- The paper is clearly structured. The algorithm is also presented clearly. However, the theoretical derivation in Section 3.1 is highly technical and hard to follow. I recommend the authors to explain the connection between the abstract variables ($a$, $b$) and the concrete diffusion process ($\epsilon_\theta$, $x_0$) earlier in the section, which would enhance clarity.

**Weaknesses:**

- The paper establishes a framework for a generic conditional generative model $M$. While the reparameterization in Section 3.2 correctly maps variables ($a\leftrightarrow \epsilon$, $b\leftrightarrow x_0$), the argument that this proves diffusion models are distribution estimators could be overstated. The theoretical framework assumes a ​linear combination​ $c_1a + c_2b$, which matches the DDPM forward process. However, the leap to the complex, non-linear reverse process learned by $\epsilon_\theta$ is not sufficiently justified. The theory does not account for the approximation error of a real-world, imperfectly trained $\epsilon_\theta$.
- The claim that the model "can effectively distinguish between ID and OOD data" in Section 3.3 is supported by an accuracy of 56.5% on ImageNet-A, only compared to a 50% random chance baseline. This is not a strong result. The paper does not compare against simple but powerful baselines like ​reconstruction error​ in the latent space of an autoencoder or other likelihood-based OOD detection methods. Without these comparisons, the practical utility of the proposed metric remains unclear.
- The results show "consistent" but often "modest" improvements. The paper should do more to qualify these gains. For instance, on text-to-image tasks, the improvements in PickScore/ImageReward are statistically significant, but what is the ​practical or perceptual significance​?

**Questions:**

- The improvements in metrics like PickScore and HPS, while consistent, are numerically small. Have you conducted any human evaluation studies (e.g., A/B tests) to determine if these metric differences correspond to a noticeable and meaningful improvement in perceived image quality or prompt alignment by human raters?

---

### Official Review · Reviewer_TxgK · 2025-10-30

**Soundness:** 1
**Presentation:** 3
**Contribution:** 2
**Rating:** 2
**Confidence:** 4

**Summary:**

The paper proposes a test-time computation strategy to improve the output of diffusion models by scaling test-time compute. Such approaches forego expensive training of the base model for more expensive queries at test time. The core of the approach is to maintain a set of candidate intermediate points in the reverse process (at the same timestep). To generate the next point in the trajectory, the paper proposes a heuristic which updates all of them and selects the one with the smallest error (as predicted by the model) as the seed for the next set of iterates. Finally, from this seed, multiple one-step denoisings are drawn to form the candidate set of the next denoising step. The authors empirically validate their findings by showing that this approach leads to improvements in generation quality for both conditional and unconditional generation.

While exploring test-time scaling approaches for diffusion models is a natural step, my main concerns are the lack of novelty and theoretical grounding for the paper. For instance, the claim by the authors at the bottom of page 3 is wrong as stated. The minimizer of $\mathbb{E} [\|M (f (a, b))\|^2]$ is $M(\cdot) \coloneqq 0$. Hence, it is not clear why $M$ would match the conditional second moment as claimed. This makes it challenging to follow the remainder of the theoretical derivation and evaluate the soundness of the approach. Another issue the choice of iterations in line 7 of the Algorithm 1. The extra noise variable $\epsilon'$ has noting to do with the iterate itself and adds seemingly unnecessary noise. Omitting it could substantially stabilize the value of this estimate. In fact, this quantity when expected over the two noise variables essentially measures how large the predicted noise is when a step from $x_{i, t}$ is taken. This formulation feels unnecessarily complex without sufficient justification. Finally, the improvements achieved by this technique seem somewhat mild in the empirical evaluation of the method. At this point, the paper still requires several improvements before acceptance.

**Strengths:**

See main review

**Weaknesses:**

See main review

**Questions:**

See main review

---

### Note · Authors · 2025-11-13

I have read and agree with the venue's withdrawal policy on behalf of myself and my co-authors.